# Atomic-resolution structure analysis inside an adaptable porous framework

Yuki Wada[1], Pavel M. Usov[1], Bun Chan ![ORCID][2], Makoto Mukaida[3], Ken Ohmori ![ORCID][1], Yoshio Ando ![ORCID][1], Haruhiko Fuwa ![ORCID][4], Hiroyoshi Ohtsu ![ORCID][1] & Masaki Kawano[1] ✉

We introduce a versatile metal-organic framework (MOF) for encapsulation and immobilization of various guests using highly ordered internal water network. The unique water-mediated entrapment mechanism is applied for structural elucidation of 14 bioactive compounds, including 3 natural product intermediates whose 3D structures are clarified. The single-crystal X-ray diffraction analysis reveals that incorporated guests are surrounded by hydrogen-bonded water networks inside the pores, which uniquely adapt to each molecule, providing clearly defined crystallographic sites. The calculations of host-solvent-guest structures show that the guests are primarily interacting with the MOF through weak dispersion forces. In contrast, the coordination and hydrogen bonds contribute less to the total stabilization energy, however, they provide highly directional point interactions, which help align the guests inside the pore.

Single-crystal X-ray diffraction (SXRD) is one of the most powerful structure elucidation tools since it can provide critical information regarding three-dimensional atom arrangement and the type of bonding involved, as well as revealing intermolecular interactions and absolute chirality. However, growing crystals of sufficiently large size for this method can be a daunting task. As a result, alternative techniques that can overcome this drawback, including electron diffraction[1] and guest encapsulation into single crystalline porous matrices, for example, the crystalline sponge (CS) method, are currently being developed[2]. One of the critical issues of the CS method is the generally unsatisfactory quality of the diffraction data, which leads to the structure models with only the molecular-level resolution[3], thus making it difficult to correctly determine the structures of complex molecules[4].

Incorporating molecules into single crystalline porous materials could reveal a plethora of important information about guest structures[2,5,6] and host-guest interactions[7–10], in addition to monitoring reaction intermediates[11,12] and detecting metastable species[13,14]. In particular, structure determination of complex organic molecules has garnered strong attention since this technique can be applied to non-

crystalline solids[15], oils[16–19], and compounds that can only be obtained in trace amounts[20–24]. Among several kinds of porous materials, metal-organic frameworks (MOFs) are typically employed as scaffolds for the encapsulation of various guests owing to the presence of large accessible pores and high designability. Frameworks that were employed for this purpose have been referred to as crystalline sponges. To date, only a handful of MOFs have been identified to be compatible with a wide range of substrates since guest capture and stabilization are typically facilitated by a series of finely balanced host-guest interactions, which are difficult to rationally design[2,3,15,25–27]. As a result, the final structures are typically only resolved down to the molecular-resolution level, that is showing the overall outline of a molecule but without pinpointing the exact positions of atoms. Fitting structure models to such data often necessitates the application of crystallographic restraints and constraints, which can significantly complicate the structure elucidation of unknown compounds[28].

Previously, we reported SXRD analysis of polycyclic aromatic hydrocarbons encapsulated into Co-3TPHAP MOF[7]. Interestingly, the host-guest crystal structures contained several solvent molecules in the pore, which were highly ordered and could be fully resolved by

[1]Department of Chemistry, School of Science, Tokyo Institute of Technology, 2-12-1 Ookayama, Meguro-ku, Tokyo 152-8550, Japan. [2]Graduate School of Engineering, Nagasaki University, Bunkyo 1-14, Nagasaki-shi, Nagasaki 852-8521, Japan. [3]Asahi Kasei Pharma Corporation, 632-1 Mifuku Izunokuni, Shizuoka 410-2321, Japan. [4]Department of Applied Chemistry, Faculty of Science and Engineering, Chuo University, 1-13-27 Kasuga, Bunkyo-ku, Tokyo 112-8551, Japan. ✉e-mail: mkawano@chem.titech.ac.jp

crystallography. They were interacting with both the host framework and aromatic guests, which resembled a solvation shell. From these results, we theorized that the encapsulating ability of Co-3TPHAP could be expanded to a wider range of organic guests and be used for the elucidation of their structures. In most MOFs used as crystalline sponges, the guests were usually captured and stabilized through direct interactions with the host framework[3]. On the other hand, the encapsulation process in Co-3TPHAP was additionally mediated by the solvent molecules forming a complex series of host-solvent-guest interacting assemblies. These kinds of secondary interactions could adapt to the guest shapes and minimize their molecular motion, thus improving crystallographic resolution.

To test this hypothesis, the encapsulation of 14 diverse organic compounds into Co-3TPHAP was investigated (Fig. 1). These were divided, based on their common structural features, into artemisinin series, steroid series, and aromatic polycyclic molecules. Moreover, three compounds that were reported recently and whose structures were characterized using only spectroscopic techniques were also analyzed using this method. The analysis of these molecules demonstrated that adaptable water networks considerably improved the structure quality (*vide infra*). The model refinement almost completely avoided any crystallographic restraints and constraints, which is a major milestone in the development of the crystalline sponge technique.

## Results and discussion

Artemisinin is a well-known drug for the treatment of malaria, with several synthetic derivatives commercially available. Their basic structures consist of sesquiterpene lactone and contain a peroxide bridge. For the purpose of exploring host-guest interactions, this series presented an opportunity to investigate the effects of relatively minor modifications to one part of the molecule on the guest distribution inside the pore of Co-3TPHAP, interactions with the internal solvent molecules and the quality of the final structure model. The chemical variations included ketone (artemisinin), ether (artemether), hydroxy (dyhydroartemisinin, DHA), and succinic acid monoester (artesunate) groups while maintaining the hydrophobic core intact.

In the artemisinin-loaded crystal, the space group changed from a centric $P2_1/n$ of the original as-synthesized Co-3TPHAP to a chiral $P2_1$, with the Flack parameter of 2.15(12)%, which was sufficiently reliable for determination of absolute chirality. Five distinct cryptographic sites were identified, one of which involved coordination to the open $Co^{2+}$ center (Fig. 2 and Supplementary Fig. 1). Every guest molecule formed at least one hydrogen bond through oxygen atoms of either the ketone group or the peroxide bridge[29]. In contrast, when the methoxy analog, artemether, was encapsulated the resultant structure contained only one unique crystallographic site. Depending on the pore content, the space group of Co-3TPHAP can interchange between $P2_1/n$ and $C2/c$, which is determined by whether the included solvent or guest molecules are placed in special positions or not[7]. In the case of artemether, the encapsulated crystal displayed a chiral $C2$ space group with the Flack parameter of 1.9(2)%.

Artesunate is the analog featuring the most dramatic modification to the core, succinate monoester attachment, which leads to markedly different properties compared to the other compounds in the series (Fig. 2). When encapsulated inside Co-3TPHAP, it formed hydrogen bonding with the internal water network through the carboxylate and peroxide groups and caused the crystal symmetry to change to a chiral $P2_1$ space group with the Flack parameter of 4.9(2)%. There were two kinds of artesunate crystallographic sites (Supplementary Fig. 2). The first featured the guest molecule coordinated to the $Co^{2+}$ ions, whereas

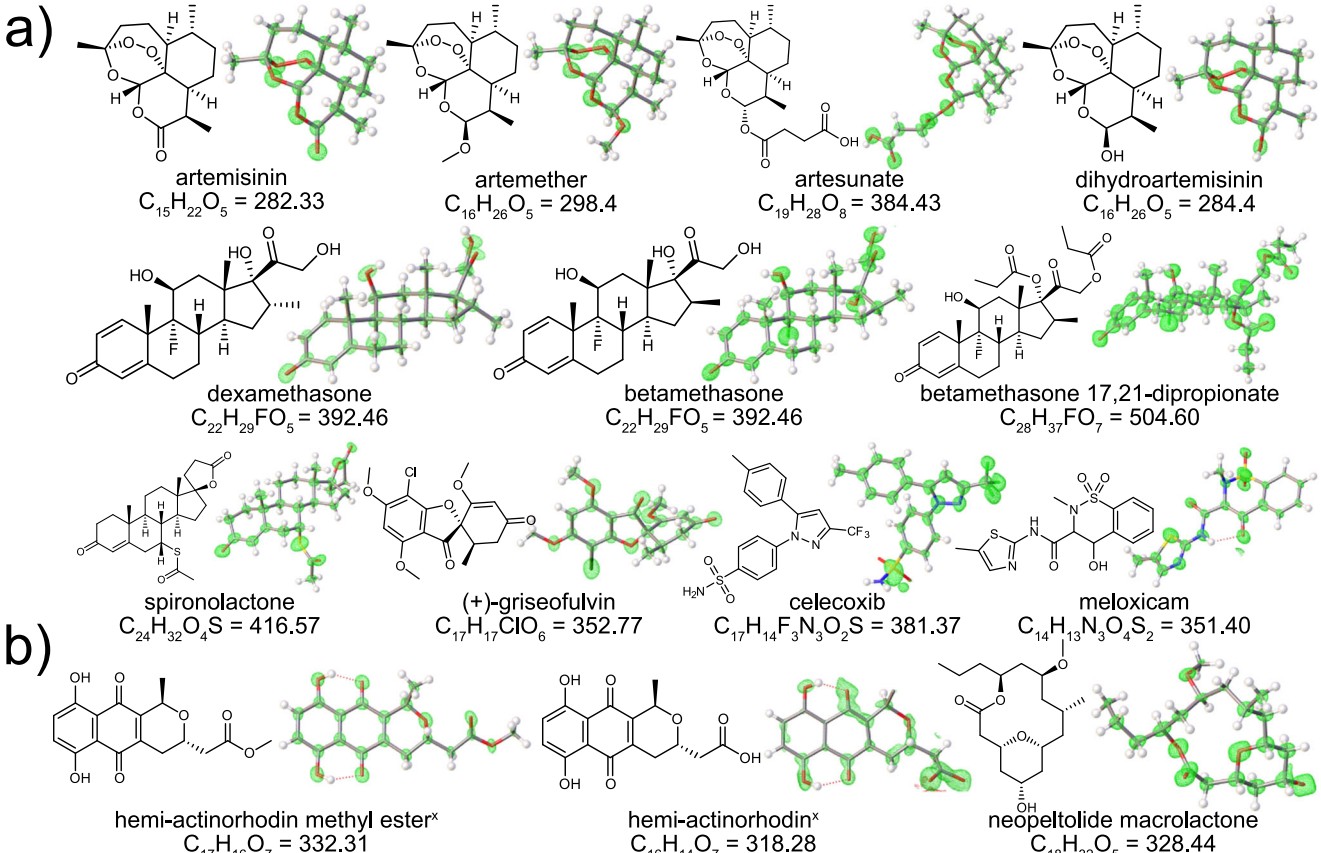

**Fig. 1 | Electron density maps (2Fo – Fc) of elucidated structures of bioactive guest compounds inside the pore of Co-3TPHAP.** The corresponding molecular formulas and molar masses (g mol⁻¹) are shown. **a** Commercially available compounds and **b** recently published molecules without available crystal structures. ˣ denotes compounds that were encapsulated as racemic mixtures.

Structures shown in figure:

artemisinin
$C_{15}H_{22}O_5 = 282.33$

artemether
$C_{16}H_{26}O_5 = 298.4$

artesunate
$C_{19}H_{28}O_8 = 384.43$

dihydroartemisinin
$C_{16}H_{26}O_5 = 284.4$

dexamethasone
$C_{22}H_{29}FO_5 = 392.46$

betamethasone
$C_{22}H_{29}FO_5 = 392.46$

betamethasone 17,21-dipropionate
$C_{28}H_{37}FO_7 = 504.60$

spironolactone
$C_{24}H_{32}O_4S = 416.57$

(+)-griseofulvin
$C_{17}H_{17}ClO_6 = 352.77$

celecoxib
$C_{17}H_{14}F_3N_3O_2S = 381.37$

meloxicam
$C_{14}H_{13}N_3O_4S_2 = 351.40$

hemi-actinorhodin methyl esterˣ
$C_{17}H_{16}O_7 = 332.31$

hemi-actinorhodinˣ
$C_{16}H_{14}O_7 = 318.28$

neopeltolide macrolactone
$C_{18}H_{32}O_5 = 328.44$

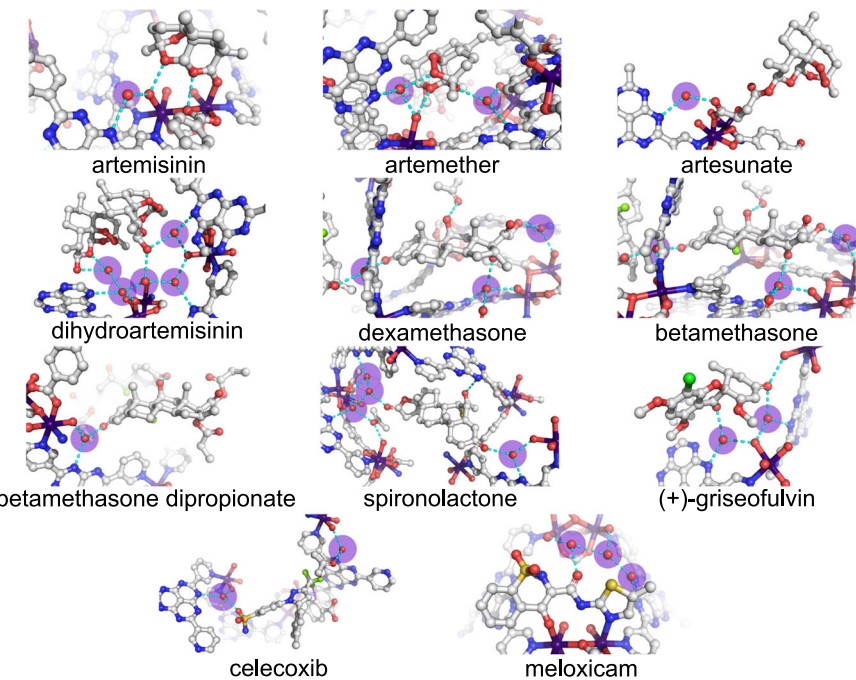

**Fig. 2 | Hydrogen bonding interactions between Co-3TPHAP, water, and each of the guests.** The water molecules highlighted in purple were freely residing inside the pore, and cyan dash lines represent hydrogen bond contacts. Atom coloring scheme: C: white, N: blue, O: red, F: yellow-green, Cl: green, S: yellow, and Co: purple.

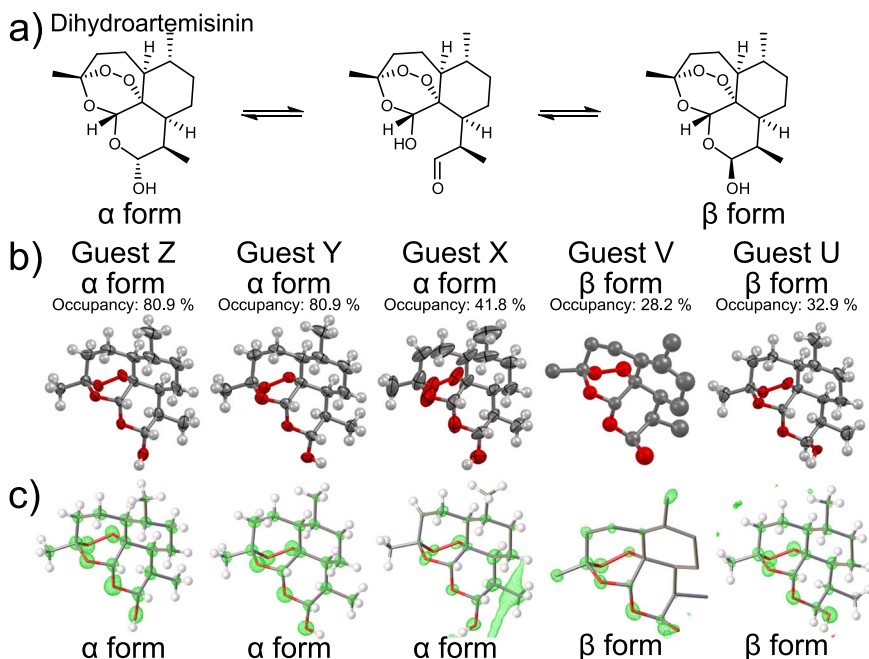

**Fig. 3 | Structural dynamics of dihydroartemisinin and its effect on the chirality of encapsulated sites. a** epimerization reaction of DHA, **b** ORTEP diagram of each site (>50% probability), and **c** the corresponding 2Fo – Fc electron density maps. The guest letters correspond to the suffix of atom labels in the CIF file. Atom coloring scheme: C: gray, N: blue, O: red, and hydrogen: white.

the second was primarily supported by hydrogen bonding with several pore water molecules. Interestingly, the carbon-oxygen bond lengths within the carboxylic acid were 1.187(7) and 1.322(7) Å for the coordinating site, and 1.245(8) and 1.309(7) Å for the hydrogen bonding site, which suggested that these groups existed in a protonated form rather than anionic.

Unlike other analogs, the hemiacetal motif in DHA can interconvert in solution to aldehyde and hydroxy groups leading to inversion of the carbon center (epimerization) in a similar fashion to carbohydrates (Fig. 3a). In solution, the epimerization reaction between the α and β forms reaches a certain equilibrium state, which depends on solvent polarity[30,31]. However, the crystal structure of only the β form has been reported to date because of more favorable crystal packing. Remarkably, after encapsulation into Co-3TPHAP both epimers were detected inside the pore. Most of the guests were observed in their α form, whereas the β form, which is predominant in the

crystalline DHA powder, accounted for only two sites with occupancies of *ca.* 30% (Fig. 3b, c). The Flack parameter of this crystal structure was 1.87(14)%, confirming the validity of the absolute configuration assignment. Since Co-3TPHAP preferentially captured the α form of DHA over the β form, it suggested that the pore environment closely resembled that of a polar solvent (Fig. 2 and Supplementary Fig. 3).

Overall, the investigation of the artemisinin series shed light on the key interactions that facilitated the structure elucidation. The presence of relatively strong directional interactions, such as coordination and hydrogen bonds, between the guests and the host framework sufficiently immobilized them inside the pore to give highly resolved electron densities with minimal disorder. Another important discovery was that minor chemical modifications of the artemisinin core resulted in dramatic changes to the location and symmetry of the observed crystallographic sites. In each case, the surrounding water network adjusted its structure to maximize the interactions with the polar groups of the guests. This kind of adaptive behavior indicated that Co-3TPHAP could be used to encapsulate guests with a diverse range of chemistries.

The next series of molecules to be investigated was based on a hydrophobic steroid scaffold, which is the basis for steroidal anti-inflammatory drugs (SAIDs). These structures can possess a high diversity of functional groups, which can have many different arrangements around the core. Therefore, this series was expected to provide a more rigorous test for the ability of the Co-3TPHAP pore to discern and adapt to different guests. The examined compounds included dexamethasone and betamethasone, which are enantiomers of each other with only a single carbon atom having different chirality. The third compound was a derivative of betamethasone with propionate ester groups at 17 and 21 positions, betamethasone dipropionate, which has a greater steric bulk and higher hydrophobicity compared to the parent molecule. Finally, encapsulation and structural analysis of spironolactone was performed, which has different functional groups all around the steroid core altering its chemical properties. The outlined series of compounds displayed an adequate variety of chemical modifications to usefully probe their effects on guest inclusion and structure visualization.

First, the dexamethasone and betamethasone encapsulated crystals showed a high degree of similarity between their host-guest structures, and the differing chiral center was clearly displayed on the electron density maps (Fig. 2; Supplementary Figs. 4 and 5). When encapsulation was performed on betamethasone dipropionate, the guest could not fit into the same space in the pore as the unsubstituted compound, and instead was positioned diagonally across (Fig. 2 and Supplementary Fig. 6). In the structure of Co-3TPHAP with spironolactone, the water shell was extended to match the molecular size of the guest providing a better fit (Fig. 2 and Supplementary Fig. 7). Overall, the hydrophobic groups did not prevent the guests from interacting with the internal water network, which was able to adapt itself and maximize the interactions. These results demonstrated that polar groups play an important role in the water-mediated encapsulation mechanism, where highly malleable water clusters were able to offer a complementary fit to the guest shapes.

The artemisinin and steroid families consisted of compounds that were primarily based on saturated sp3-hybridized hydrocarbon skeletons. Therefore, to improve molecular diversity, the third series was selected to feature more planar aromatic moieties with many polar heteroatom-based groups. The molecules in this series were not analogs derived from a single scaffold but were rather selected with a focus on diversifying their shapes, sizes, and chemical functionalities. These included (+)-griseofulvin, meloxicam, and celecoxib, all of which are commercially available drugs. Once encapsulated inside Co-3TPHAP, all the guests could be clearly visualized inside the pore and displayed interactions with the water molecules via hydrogen bonds (Fig. 2 and Supplementary Figs. 8-10). It is important to note that since

these compounds had generally poor solubility in pure heptane, and acetone (10% v/v) was added to the encapsulation solutions (Supplementary Tab. 1).

To gain deeper insights into the solvent effects, the solubility of each compound at 40 °C was determined using ultra-performance liquid chromatography-mass spectrometry (UPMS). The results showed that the guest concentrations varied dramatically among the compounds (Supplementary Tab. 2). In general, the addition of 10% of acetone improved the solubility by one or two orders of magnitude. Remarkably, some guests (dexamethasone, betamethasone, and meloxicam) exhibited solubilities in the µg mL$^{-1}$ range, indicating that they could still be captured by Co-3TPHAP even from highly dilute solutions. Meloxicam presented an interesting case study on the interplay between guest solubility and data quality. When inside the framework, it acted as a bidentate ligand coordinating with two cobalt ions in the same cluster. Therefore, in pure heptane, the equilibrium was strongly favored towards the incorporation of meloxicam into Co-3TPHAP. In comparison, when acetone was added, the electron density in the resultant crystal was less resolved, indicating that the guest occupancy was lower even though the amount of available meloxicam in solution should have been doubled.

To rationalize this behavior, it is necessary to consider a competition between polar solvent molecules and a guest for the occupation of interactive sites inside the framework pore. In highly non-polar solvents, such as heptane, the guests are more likely to form the strongest interactions with the framework and be more visually resolved by SXRD. However, the addition of more interactive solvents could disrupt this process and displace the guest molecules from their preferred crystallographic sites. The case of meloxicam is quite unique since several other compounds in this study were encapsulated with the help of acetone and showed improved diffraction. In fact, in some of the final structures, acetone molecules were interacting with the guests and providing additional stabilization. These results highlight the complexity of host-solvent-guest interactions and demonstrate the difficulty of finding universally applicable encapsulation conditions.

It was previously suggested that molecular structures obtained using the crystalline sponge method cannot provide high precision for structural parameters, such as bond lengths and angles due to lower resolution[3]. In contrast, the X-ray diffraction spots in Co-3TPHAP crystals with encapsulated guests could be observed at exceptionally high angles for a porous framework. The presence of highly ordered internal water networks was postulated to be primarily responsible for this behavior, which allowed to reveal the guest structures with atomic resolution. To further validate the quality of the obtained guest crystal structures, four of the drug molecules used in the encapsulation study were crystallized by conventional solvent evaporation methods. SXRD measurements were performed using the same experimental setup as for Co-3TPHAP. A comparison of bond lengths in both crystals showed that the differences were generally negligible, and the corresponding errors were in a similar range, especially when the guest occupancy was above 50% (Supplementary Figs. 14–18). Since the lower occupancy (<50%) guest models were more strongly affected by residual electron density from the disorder partners, some bond lengths deviated significantly out of the normal range. Additionally, the more significant differences in some individual bonds could be explained by the differing surrounding environments and interactions since the guests inside Co-3TPHAP were also subjected to additional solvation effects. The refined Flack parameters were also found to be closely matching, however in the case of encapsulated guests, they could be determined to a greater number of decimal places due to the presence of heavy cobalt ions.

Co-3TPHAP displayed an excellent performance in elucidating molecular structures of various bioactive compounds, however, they have already been studied in great detail in the past and can readily form

crystals using conventional approaches. To further expand the scope of this technique, structural analysis of less characterized natural products and intermediates was also undertaken (Fig. 2 and Supplementary Figs. 11–13). The first two compounds were hemi-actinorhodin and its methyl ester, which are total synthesis intermediates of actinorhodin, a polyketide antibiotic produced by Streptomyces[32,33]. These compounds were encapsulated from racemic mixtures because of limited sample availability, which precluded chiral separation[34].

The crystal structure of the methyl ester analog showed five crystallographic sites, each interacting with the coordinated and hydrogen-bonded water molecules (Fig. 2 and Supplementary Fig. 11). On the other hand, the carboxylic acid compound, unexpectedly, displaced terephthalate co-ligand from the framework backbone and chelated to the cobalt dimer (Fig. 2 and Supplementary Fig. 12). Interestingly, the existence of two strong coordination bonds proved detrimental to the electron density resolution of the guest with the final structure model requiring the use of restraints to provide adequate statistics (Supplementary Tab. 3). This result suggests that increasing host-guest interaction strength might not always lead to reduced crystallographic disorder since the framework could become less flexible in adapting to specific guest features. Both compounds were solved in a chiral space group and the Flack parameter was almost 50%, which meant that the encapsulated crystals were almost fully racemized. This phenomenon provided a clue to the origin of the chiral recognition ability of Co-3TPHAP.

While the as-synthesized framework crystallized in an achiral space group, its cobalt dimer has a non-superimposable mirror image (Fig. 4a). However, within a unit cell, both enantiomers exist in equal numbers. Each cluster of the same chirality was bridged by the ligands creating two chiral pores. As a result, both enantiomers of hemi-actinorhodin guests could be selectively captured into their preferred chirality pores. This behavior caused the final encapsulated MOF structures to retain their racemic nature and give Flack parameter values of around 50%, which is consistent with opposite mirror images canceling each other. This phenomenon was observed during the encapsulation of both racemic compounds, as well as in other studies using different MOFs[15,35], indicating that it could be a more general trend.

The final compound analyzed using Co-3TPHAP was neopeltolide macrolactone[36], which is the core of a natural compound (+)-neopeltolide. It was originally isolated by Wright and co-workers from a Jamaican deep-water sponge that belongs to the family Neopeltidae[37]. Since this compound exists in an oil state at ambient temperature due to the flexibility of the macrocycle, SXRD analysis of this derivative has not yet been reported. The absolute structure was proposed by extensive 2D NMR analysis, and the relative configuration was assigned based on NOESY correlations. However, further research proposed the reassignment of its stereochemistry by total synthesis[38,39]. Encapsulation of neopeltolide macrolactone into Co-3TPHAP clearly revealed its atomic coordinates allowing unambiguous assignment of absolute chirality, thus significantly expediting the analysis of its three-dimensional structure (Fig. 2 and Supplementary Fig. 13).

The driving forces behind the unique encapsulation phenomena inside Co-3TPHAP were investigated by computational modeling (Supplementary Tab. 4). The energy calculations of the host-guest structures showed that the guests were predominantly interacting with the MOF through long-range dispersion forces. The coordination and hydrogen bonds complement these by providing highly directional point interactions, which helped to align the guests inside the pore allowing them to achieve greater crystallographic resolution.

In summary, the structures of a wide range of guests were successfully analyzed by the crystalline sponge method down to atomic-level resolution. The key feature that facilitated this analysis was the presence of adaptable water networks in the pores that could selectively capture each guest and significantly improve their molecular alignment. The participation of highly directional hydrogen bonds was especially crucial for this process. Furthermore, the hydrophobic space existing in the center of the pore could effectively accommodate non-polar groups, which was complementary to the hydrophilic pore walls.

The resultant host-guest structures exhibited a remarkable level of diffraction data quality, considering the large volume of the pore, allowing to clearly resolve positions of every atom. As a result, the structure models almost completely avoided the use of any restraints or constraints during the refinement, which is particularly advantageous for the structure elucidation of unknown compounds. The scope and depth of the presented results place Co-3TPHAP at the

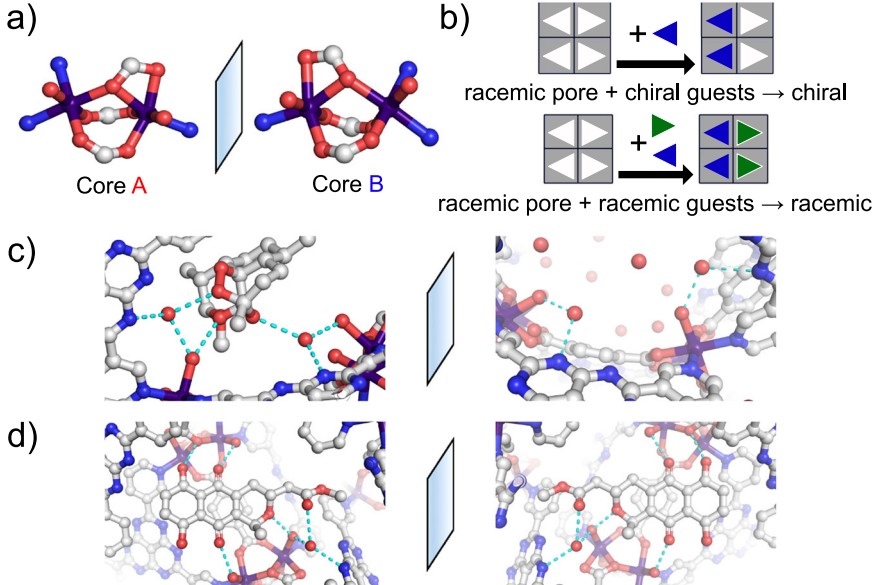

**Fig. 4 | The origin of chirality recognition in Co-3TPHAP. a** Two enantiomeric metal nodes in Co-3TPHAP, **b** a schematic illustration of chirality recognition. A pair of enantiomeric pores with **c** chirally pure artemether, and **d** racemic hemi-actinorhodin methyl ester. Atom coloring scheme: C: white, N: blue, O: red, and Co: purple, hydrogen atoms were omitted for clarity.

forefront of crystalline sponge frameworks. This remarkable performance was made possible by multiple coexisting structural motifs working in concert inside the pore. These features could be further refined and improved in the next iteration of material design enhancing its scope even further and making it competitive with traditional structural analysis techniques.

## Methods
### Chemicals
Solvents and reagents were purchased from KANTO CHEMICAL CO., INC., TCI Co., Ltd., FUJIFILM Wako Pure Chemical Corporation and used without further purification except where noted.

### Single-crystal growth of guest molecules
Guest single crystals were obtained by the slow evaporation method. The solvents used were 1 mL $n$-hexane for artemisinin, 1 mL acetone for dexamethasone, betamethasone dipropionate and (+)-griseofulvin, and 2 mL acetone + 1 mL ethanol for meloxicam. The guest solutions were placed into 4 mL vials and a needle was inserted into the caps. Single crystals were obtained after several days by slow evaporation of the resultant solutions in an incubator at 14 °C.

### Encapsulation experiments
Co-3TPHAP single crystals immersed in ethyl acetate were transferred to an empty glass vial. Fresh $n$-heptane was added and then kept for 1 min before the liquid was decanted, leaving the crystals behind. The process was repeated three times to get Co-3TPHAP exchanged with $n$-heptane. The guest powders were weighed into separate 2 mL vials, followed by the addition of the encapsulation solvent (Supplementary Tab. 1). After that, a drop of $n$-heptane containing a few Co-3TPHAP crystals was transferred to each guest solution vial. They were then tightly capped and placed into an incubator at 40 °C. After 3 days, the loaded crystals were analyzed by single-crystal X-ray diffraction without any additional treatment.

### General methods and single-crystal X-ray analysis instrumentation
The X-ray analysis was performed on a diffractometer equipped with a beamline BL-5A at KEK (the High Energy Accelerator Research Organization, Japan) with a Pilatus3 S6M detector (synchrotron, $\lambda = 0.7500$ Å, $T = 95$ K). XDS software was used for the processing and data reduction. The structures were solved by Dual space methods (SHELXT-2018) and refined by full-matrix least-squares calculations on $F^2$ (SHELXL-2018) using the OLEX program package. All non-hydrogen atoms were refined with anisotropic displacement parameters. All hydrogen atoms were created with ideal geometry and refined using a riding model. For all structure analyses, the carbon or oxygen atoms with the Q suffix correspond to the electron density for which models could not be reasonably determined. We assumed that the unassigned electron density was due to $n$-heptane or water inside the pore. The type (isotropic or anisotropic) of atomic displacement parameter refinement for each guest site depended on their occupancy and the overall degree of disorder. Generally, models with less than 50% occupancy were strongly affected by the residual electron density of the disordered partners, such as solvents. The occupancy of these partners was often even lower than the guests themselves due to the diffuse low electron density appearance. As a result, some of the low occupancy sites were refined as isotropic models.

### Solubility measurements
Solubility was measured in a slurry in pure $n$-heptane or 10 % acetone in heptane. Samples were placed into vials and shaken in a MyBL-100CS shaker (ASONE, Osaka, Japan) at 40 °C for 18 h. The resultant suspensions were filtered through a 0.45 μm PVDF centrifugal filter. Analysis was performed by ultra-performance liquid chromatography-MS (UPLC-MS) on an Acquity Premier UPLC system (Waters Corporation, MA, USA), equipped with a PDA detector, SQD2 detector, and a 2.1 × 50 mm Acquity UPLV BEH C18 column (Waters Corporation, MA, USA). A gradient method from 5% to 95% acetonitrile in 10 mM ammonium acetate aqueous solution at 0.6 mL/min for 0.6 min was used. The mass spectrometer was operated with an electrospray source. Dexamethasone, betamethasone, and artesunate dissolved in pure $n$-heptane were analyzed in a positive ionization mode. The other solutions were analyzed using the PDA detector.

### Host−Water−Guest interaction calculation
First, to create initial files for calculation, a set of cif files for each crystallographic site was created, and the water molecules interacting with the guest were identified and the other pore contents were deleted. Since the location of hydrogen atoms cannot be determined by single-crystal X-ray diffraction alone, the location and orientation of water in pores were optimized. The binding energy with the MOF and non-covalently bound water was then calculated using the XTB2 method; the binding energies were normalized on a per-guest molecule basis. The per-guest binding energy between MOF-guest (A) and guest-water (B) was also calculated separately. We found that the difference between (A) + (B) and the sum of actual binding energy was comparatively small.

## Data availability
All data are available in the main text or the supplementary materials. The X-ray crystallographic data for structures reported in this article have been deposited at the Cambridge Crystallographic Data Center (CCDC) under accession numbers CCDC 2246417 (hemiactinorhodin methyl ester@Co-3TPHAP), 2246418 (hemi-actinorhodin@Co-3TPHAP), 2246419 (neopeltolide macrolactone@Co-3TPHAP), 2246420 (artemisinin), 2246421 (dexamethasone), 2246422 (betamethasone dipropionate), 2246423 (meloxicam), 2246424 ((+)-griseofulvin), 2246425 (artemisinin@Co-3TPHAP), 2246426 (artemether@Co-3TPHAP), 2246427 (artesunate@Co-3TPHAP), 2246428 (dihydroartemisinin@Co-3TPHAP), 2246429 (dexamethasone@Co-3TPHAP), 2246430 (betamethasone@Co-3TPHAP), 2246431 (betamethasone dipropionate@Co-3TPHAP), 2246432 (spironolactone@Co-3TPHAP), 2246433 ((+)-griseofulvin@Co-3TPHAP), 2246434 (celecoxib@Co-3TPHAP), 2246435 (meloxicam@Co-3TPHAP), which can be obtained from the CCDC via https://www.ccdc.cam.ac.uk/structures/. All other data are available from the corresponding author upon request.

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

## Acknowledgements

This research was supported by JST SPRING, Grant Number JPMJSP2106 (Y.W.) JSPS KAKENHI Grants JP20H04662 (M.K.), JP21K18976 (M.K.), the JSPS Grant-in-Aid for Scientific Research on Innovative Areas, Materials Science of Meso-Hierarchy JP23H04878 (M.K.), the JSPS A3 Foresight Program, Asahi Kasei Pharma Corp, Hyogo Science and Technology Association. The X-ray crystallographic analysis was performed under the approval of the Photon Factory Program Advisory Committee (Proposal No. 2021G046). We thank Dr. Chiaki Tsuboi (Asahi Kasei Pharma Corp.) for the helpful discussion and Ms. Satoko Katsumata (Asahi Kasei Pharma Corp.) for ultra-performance liquid chromatography-MS (UPMS). Computational resources were partly provided by the RIKEN Information Systems Division (Q22266).

## Author contributions

Y.W.: experiments, analysis, and writing of the manuscript. P.M.U.: supervision, discussion, and editing of manuscript. B.C.: calculation. M.M.: experiment (meloxicam encapsulation, UPMS measurement). K.O.: sample preparation (hemi-actinorhodin and its methyl ester), Y.A.: sample preparation (hemi-actinorhodin and its methyl ester), H.F.: sample preparation (neopeltolide macrolactone), H.O.: supervision, discussion. M.K.: conceptualization, funding acquisition, supervision, and editing of the manuscript.

## Competing interests

The authors declare no competing interests.
