## [Peer Review File · Nature Communications]

Atomic-resolution structure analysis inside an adaptable porous frameworkReviewers' Comments:

Reviewer #1:

Remarks to the Author:

This work further developed CS methods for the determination of molecular structures. With only one MOF, a few complicated molecules were determined, including two for the first time. The overall work is done well and also important for broad audience. I recommend the acceptance after addressing following question.

I guessed the authors tried more guests and for those guests which is not included in the paper, the structure might be not ordered enough. Can they explain how to identify in the beginning which is good guest for such a MOF?

If mixed organic molecules in the solution, will it possible to encapsulate them in different crystals?

ADPs are significantly large for the MOF-guest structures. What is the reason? How about the ADPs for the MOF itself?

in Celecoxib@Co-3TPHAP, the guest molecules are not in a proper shape, such as the C-N bonds.

in Hemi-actinorhodin methyl ester@Co-3TPHAP, one C has much large ADP than its neighbouring atoms.

I didn't see any structure with the space group of C2/c.

In some refinement, there are some many omitted reflections. Is there any special reason?

Reviewer #2:

Remarks to the Author:

The manuscript titled 'Atomic-resolution structure analysis inside an adaptable porous framework' by Masaki Kawano and co-workers reported an alternative crystalline sponge Co-3TPHAP for structure determination of 14 bioactive compounds. In this work all the encapsulated compounds were successfully determined. Co-3TPHAP demonstrated a versatile crystalline sponge and entrapped the guest molecules via hydrogen bonding and co-coordinative alignment and encapsulated hydrophilic and hydrophobic guests in its pores.

The field of crystalline sponge method is still developing and development of an alternate crystalline sponges is a significant contribution to the field. However, crystallography is the major part of crystalline sponge method. Refining and modelling of the guest structures is important in structure determination. In this paper the crystallography of the guest molecules were not executed properly. Therefore, I recommend not to publish in Nature Communication.

Comments to authors:

1. Most of the guest molecules were refined isotropically and authors did not explain anywhere in the supporting information about keeping guests molecules isotropic.

For example structure 2246417 hemi-actinorhodin methyl ester @Co-3TPHAP when I refined all the non-hydrogen atom anisotropically the R value rises to 15% which is not acceptable for structure determination. Also, it would not support the authors claim of 'resolution down to atomic level'.

2. Several guest molecules have lower occupancy below 30% and were also refined isotropically. Authors did not attempt to explain the lower occupancy and any attempts they made to improve the occupancy.

3. Dexamethasone C-C bond length varies from 1.34 to 1.6 Å which is not good practice for structure determination and also have large ellipsoids.

4. Points 1-3 are very commonly observed in crystalline sponge method and I am not convinced that how this sponge has better resolution than any other crystalline sponge published before. To make their argument better authors should have compared their finding with other published work.

5. There were several MOFs were used as successful crystalline sponges authors mention in the manuscript ' To date only a handful of MOFs have been identified to be compatible for a wide range of substrates' but did not cite other crystalline sponges.

6. Authors did not cite reviews written on crystalline sponge methods.

ACS cent. Sci., (2021), 7, 404-414.

Angew.Chem.Int.Ed., (2021), 60, 25204-25222.

Materials Today: Proceedings, (2022), 56, 3766-3773.

Reviewer #3:

Remarks to the Author:

The authors have obtained a series of interesting crystal structures of host-guest combinations in this paper, but the ideal is not very novel, and the strategy used is also the strategy of many published papers before. Personally, this is actually a very classical host-guest chemistry, through the crystallization method to obtain some host-guest combined crystal structure, not suitable for the concept of atomic resolution. There is a suspicion of grandstanding. It is suggested that the author should readjust his thinking and condense the innovative ideas of the paper. So I do not recommend publishing in this journal with the present manuscript. It is recommended that the authors provide the powder diffractions of these host-guest compounds to demonstrate that a large amount of samples can be obtained. At the same time, there are some writing errors in the article, such as the use of italics in the writing of space groups. Please provide the revised documents.

Response to Reviewer's Comments for the Manuscript NCOMMS-23-25075-T:

Reviewer/Editor Comments to the Authors are in *black*, and our responses are in *blue*.

REVIEWER COMMENTS

Reviewer #1 (Remarks to the Author):

This work further developed CS methods for the determination of molecular structures. With only one MOF, a few complicated molecules were determined, including two for the first time. The overall work is done well and also important for broad audience. I recommend the acceptance after addressing following question.

We thank the reviewer for their kind words and their willingness to accept our manuscript for publication.

I guessed the authors tried more guests and for those guests which is not included in the paper, the structure might be not ordered enough. Can they explain how to identify in the beginning which is good guest for such a MOF ?

We have indeed tried to encapsulate several other molecules into Co-3TPHAP, however their structures could not be resolved. The success of the crystalline sponge technique relies on two factors: the amount of adsorbed guest inside the framework and their ordering inside the pore. Spectroscopic measurements, such as infrared spectroscopy, could reveal the presence of the encapsulated molecules, however they provide only limited information about their positioning and alignment. Because of this limitation, the X-ray diffraction techniques still remain the surest way to judge if a molecule's structure can be visualized or not. Powder X-ray diffraction (PXRD) could be used to obtain the necessary information and the data could be collected faster than for the single crystal diffraction but it usually requires larger amount of MOF sample that needs to be physically separated from the guest encapsulation mixture. Furthermore, to prevent the loss of solvent from the pores, which could negatively affect crystallinity, specialized sample preparation methods would be required. Therefore, in our opinion, collecting diffraction images on a single crystal is the most useful strategy for judging the success of guest encapsulation. Looking at several statistical parameters during the initial structure refinement, such as R_{int} and Flack parameter, is usually sufficient to decide if a particular diffraction dataset is worth refining further to obtain the guest structure.

In addition, we are aiming to employ computational modelling of host-guest structures to identify the preferential interaction modes. This information could be used to design new crystalline sponges with improved affinity for the desired guest molecules. However, such systems are very complex to model accurately since the pore environment, including the number of solvents and the presence of highly directional interactive sites could have a dramatic effect on the guest conformation and its location inside the pore.

If mixed organic molecules in the solution, will it possible to encapsulate them in different crystals?

One example of mixture encapsulation that was investigated in the past is racemic mixtures containing the opposite enantiomers. In such cases, if the crystalline sponge does not possess an inherent chirality, both enantiomers are usually included into the pore (*Cryst. Growth Des.* **18**, 126–132 (2018), *Chem* **9**, 227–241 (2023)). In another example, encapsulation of several molecules at the same time inside a metal-macrocylic framework was investigated, with each guest located at different interactive sites within the pore (*Angew. Chem. Int. Ed.* **53**, 8310–8315 (2014)). From these cases, it is reasonable to assume that a framework could

capture multiples molecules from a mixture solution with the ratio dependent on each individual affinity. Regarding encapsulation of different mixture components into different crystals, the procedure used in the present work involved soaking a few Co-3TPHAP crystals in solutions containing an excess of target guests for several days. It is expected that under such conditions a close to equilibrium state would be reached, which means that identical sites in each framework crystal would be occupied equally. Therefore, the pore contents of all encapsulated crystals would be very similar. It might be possible to design an encapsulation experiment where guest amount and their distribution within the framework crystals will not be homogeneous, for example by creating guest concentration gradients, however, such experiments lie outside the scope of the current manuscript.

ADPs are significantly large for the MOF-guest structures. What is the reason? How about the ADPs for the MOF itself?

The MOF crystal structures are different from small molecule structures due to the presence of large pore space filled with movable solvent molecules, which imparts greater flexibility and distortion. Because of this effect, the atomic displacement parameters (ADP) in MOFs tend to be larger and can be further worsened by the presence of disorder and defects. In Co-3TPHAP, the structural distortions mostly arise from the wagging motion of 3-pyridyl groups and flexibility of cobalt coordination sphere (*Inorg. Chem.* **60**, 17858–17864 (2021)). After guest encapsulation, the structure of framework experienced significant modification to the unit cell parameters indicating the presence of additional guest-induced internal strain. Despite these factors the ADP values of molecules encapsulated inside Co-3TPHAP are among the best compared to other MOFs used in the crystalline sponge application. This is especially evident in the reported ORTEP figures, which usually display only 30% probability (*IUCrJ* **3**, 139–151 (2016)). In the case of Co-3TPHAP models, we were able to obtain reasonable ellipsoids at 50% probability.

in Celecoxib@Co-3TPHAP, the guest molecules are not in a proper shape, such as the C-N bonds.

To address this comment, the Celecoxib@Co-3TPHAP was refined more to improve the structure model. We assume that the reviewer meant the C-N bonds around N3LD. This atom was affected by the residual electron density that was not modelled in the submitted structure. Therefore, to improve the model quality, the Q peak corresponding to that electron density was assigned to water and refined for a few cycles. The deposited CIF file and the crystallographic table were updated as follows:

Table S5. Crystallographic table in supplementary materials

Empirical formula: $C_{101.11}H_{53.42}Co_4F_{2.08}N_{20.38}O_{27.51}S_{0.69}$ was changed to $C_{101.11}H_{53.82}Co_4F_{2.07}N_{20.38}O_{27.09}S_{0.69}$

Formula weight: 2291.28 was changed to 2293.82

Data/restraints/parameters: 47129/2/1703 was changed to 47129/1/1705

Goodness-of-fit on F^2 : 1.035 was changed to 1.03640

Final R indexes [$I \geq 2\sigma(I)$]: $R_1 = 0.0815$, $wR_2 = 0.2512$ was changed to $R_1 = 0.0815$, $wR_2 = 0.2522$

Final R indexes [all data]: $R_1 = 0.1068$, $wR_2 = 0.2797$ was changed to $R_1 = 0.1067$, $wR_2 = 0.2808$

Largest diff. peak/hole / $e \text{ \AA}^{-3}$: 0.89/-1.07 was changed to 0.89/-1.07

in Hemi-actinorhodin methyl ester@Co-3TPHAP, one C has much large ADP than its neighbouring atoms.

The hemi-actinorhodin methyl ester molecules in question had much lower occupancy (guest U: 38.2% and guest V: 44.7%), which made them difficult to model properly since they were strongly affected by the residual electron density from solvent and the presence of inherent disorder (Figure R1). Because the residual electron density of sites U and V was only $0.3 e \text{ \AA}^{-3}$, the identification of an appropriate disorder model was too

challenging. As a result, the ADP of one of the carbon atoms was unexpectedly larger compared to neighboring atoms. To improve the guest model, the structure was refined further, and additional constraints were applied.

Figure R1. The difference density map ($0.3 \text{ e}\text{\AA}^{-3}$ level) of hemi-actinorhodin methyl ester@Co-3TPHAP. a) guest U, and b) guest V.

To include this new refinement, the deposited CIF file and the crystallographic table were updated as follows:

Table S3. A list of crystallographic restraints and constraints applied during the structure analysis of encapsulated guest in the supplementary materials.

Hemi-actinorhodin methyl ester Guest V

EADP O18V C20V C21V was added to the table.

Hemi-actinorhodin methyl ester Guest U

SIMU and DELU (O18U > C21U) was changed to EADP O18U C20U C21U

Table S5. Crystallographic table in the supplementary materials.

Empirical formula: $\text{C}_{103.35}\text{H}_{56.57}\text{Co}_4\text{N}_{18}\text{O}_{38.75}$ was changed to $\text{C}_{103.25}\text{H}_{56.98}\text{Co}_4\text{N}_{18}\text{O}_{31.73}$

Formula weight: 2406.22 was changed to 2404.97

Reflections collected: 311707 was changed to 311690

Independent reflections: 90673 [$R_{\text{int}} = 0.0142$, $R_{\text{sigma}} = 0.0128$] was changed to 90667 [$R_{\text{int}} = 0.0142$, $R_{\text{sigma}} = 0.0128$]

Data/restraints/parameters: 90673/264/3138 was changed to 90667/339/3134

Goodness-of-fit on F^2 : 1.077 was changed to 1.072

Final R indexes [$I \geq 2\sigma(I)$]: $R_1 = 0.0533$, $wR_2 = 0.1681$ was changed to $R_1 = 0.0531$, $wR_2 = 0.1671$

Final R indexes [all data]: $R_1 = 0.0557$, $wR_2 = 0.1735$ was changed to $R_1 = 0.0556$, $wR_2 = 0.1725$

Largest diff. peak/hole / $\text{e}\text{\AA}^{-3}$: 0.94/-0.69 was changed to 0.94/-0.73

Flack parameter: 0.4959(14) was changed to 0.4957(1)

I didn't see any structure with the space group of $C2/c$.

In the original paper that reported synthesis and characterization of Co-3TPHAP, *Inorg. Chem.* **60**, 17858–17864 (2021), one of the crystal structures (CCDC number 2000902) had a space group of $C2/c$. To clarify that the space group change was reported in the previous paper, a citation was added to the following sentence:

Page 3 paragraph 1: “Depending on the pore content, the space group of Co-3TPHAP can interchange between $P2_1/n$ and $C2/c$, which is determined by whether the included solvent or guest molecules are placed on special positions or not.” was changed to “Depending on the pore content, the space group of Co-3TPHAP can interchange between $P2_1/n$ and $C2/c$, which is determined by whether the included solvent or guest molecules are placed on special positions or not⁷.”

In some refinement, there are some many omitted reflections. Is there any special reason?

Since XDS software that was used to integrate the collected diffraction images does not have a beam stop blind as a basic function, the omit operation was applied instead to avoid the effects of the primary X-ray beam.

Reviewer #2 (Remarks to the Author):

The manuscript titled ‘Atomic-resolution structure analysis inside an adaptable porous framework’ by Masaki Kawano and co-workers reported an alternative crystalline sponge Co-3TPHAP for structure determination of 14 bioactive compounds. In this work all the encapsulated compounds were successfully determined. Co-3TPHAP demonstrated a versatile crystalline sponge and entrapped the guest molecules via hydrogen bonding and co-coordinative alignment and encapsulated hydrophilic and hydrophobic guests in its pores.

The field of crystalline sponge method is still developing and development of an alternate crystalline sponges is a significant contribution to the field. However, crystallography is the major part of crystalline sponge method. Refining and modelling of the guest structures is important in structure determination. In this paper the crystallography of the guest molecules were not executed properly. Therefore, I recommend not to publish in Nature Communication.

Comments to authors:

1. Most of the guest molecules were refined isotropically and authors did not explain anywhere in the supporting information about keeping guests molecules isotropic.

For example structure 2246417 hemi-actinorhodin methyl ester @Co-3TPHAP when I refined all the non-hydrogen atom anisotropically the R value rises to 15% which is not acceptable for structure determination. Also, it would not support the authors claim of ‘resolution down to atomic level’.

Since this study deals with complex host-guest crystal systems, the refinement strategy should be considered on a case-by-case basis. In particular, guest sites with occupancies below 50% can be strongly affected from the residual electron density of a disordered partner, such as solvent molecules. This kind of disordered solvent typically has even lower occupancy due to their diffuse low electron density appearance. Anisotropic refinement of such low occupancy guests would constitute an unnecessary overfitting of the structure model with questionable basis in reality. As a result, the models were refined isotropically to avoid the effects from the residual solvent electron density. We believe that there are sufficient reasons to justify this operation. Isotropic non-hydrogen atom models in single crystal analysis are perfectly acceptable in special circumstances. Additionally, we cannot know for certain why the R factor increased from 5% to 15% after the reviewer’s attempt at refining the hemi-actinorhodin methyl ester@Co-3TPHAP structure. However, at first

glance, such an increase implies an improper model, which does not accurately represent the host-guest structure. Based on the R factors reported in the current manuscript, we stand firmly behind our crystal structure refinement approach and believe that our structure models provide the best solutions to the collected diffraction data.

Nevertheless, we admit that there are insufficient details regarding the refinement process provided in the manuscript. Accordingly, the following sentences were added to the “General methods and single crystal X-ray analysis instrumentation” section in the main text:

“The type (isotropic or anisotropic) of atomic displacement parameter refinement for each guest site depended on their occupancy and the overall degree of disorder. Generally, models with less than 50% occupancy were strongly affected from residual electron density of the disordered partners, such as solvents. The occupancy of these partners was often even lower than the guests themselves due to the diffuse low electron density appearance. As a result, some of the low occupancy sites were refined as isotropic models.”

Regarding the atomic resolution, the definition we employed in the present manuscript was – individual atoms are clearly visible, and an accurate three-dimensional model can be constructed with minimal restraints and constraints. That means that the observed electron densities should be localized and perfectly superimposed with the expected atomic positions within a molecule. If we compare our guest structures to those obtained by other crystalline sponges, the difference in resolution of electron densities becomes readily apparent (Figure R2). This resolution level holds true for all the high occupancy sites of all guest molecules encapsulated inside Co-3TPHAP. Therefore, we believe that the use of “atomic resolution” terminology is fully appropriate in the present work. The matter of contention then comes down to the low occupancy sites, where the resolution of some guests indeed could not be categorized as “down to atomic level”. However, it is important to consider the goals of the current study specifically and the field of crystalline sponge development in general. That is to obtain the structures of the target molecules with high quality and as unambiguous as possible assignment of atoms and absolute configuration by encapsulation into porous crystalline matrices. As with all the crystallography, it is important to strive to refine all the atoms in the model as best as can be reasonably achieved. However, in situations where several guest sites are present inside a crystalline sponge, the complete structure model from only one site would be sufficient to fully reveal the target molecule structure. In such cases, we believe that the crystalline sponge analysis can still be considered a complete success even if the models of other guest sites do not meet the atomic resolution level standard. In this context, the quality of organic molecules structures obtained by encapsulation inside Co-3TPHAP is consistently among the best across the board (as can be seen in Figure 1 in the main text) in the field of crystalline sponge analysis.

Figure R2. The electron density maps of two guests resolved in different crystalline sponges. a) Electron density map ($2F_o - F_c$; $3 \text{ e}\text{\AA}^{-3}$) superimposed on an artemether guest inside Co-3TPHAP. b) Electron density map ($2F_o - F_c$; contour: 0.5) superimposed on a guaiazulene guest inside ZnI_2 -TPT crystalline sponge (Supplementary Fig. 1b. from *Nature* **501**, 262–262 (2013)).

2. Several guest molecules have lower occupancy below 30% and were also refined isotropically. Authors did not attempt to explain the lower occupancy and any attempts they made to improve the occupancy.

The response to the reviewer's first comment addresses the reason behind the isotropic refinement of low occupancy guests.

In relation to the improvement of guest occupancy, while it would be desirable to achieve a near 100% occupancy for all the sites inside the crystalline sponge, it is an unreasonable and an unnecessary requirement. In a framework with a large pore space, such as Co-3TPHAP, there will be many locations that a guest can occupy. However, each of these sites would have a different affinity for a particular molecule, which means that under the standard encapsulation conditions at room temperature they would not be occupied equally. It might be possible to employ harsher encapsulation conditions, for example higher temperature, to force the guest to fill all the available space. But such treatment could be detrimental to the guest compounds with sensitive parts, often found in many natural products, which would lead to their decomposition before encapsulation is completed. In addition, as mentioned in the response to the first comment, having at least one crystallographic site with more than 50% occupancy is usually sufficient for solving the structure of a target guest molecule. Overall, attempting to increase the occupancy of all the sites inside a crystalline sponge is not always a productive use of time, especially if it increases the risk of guest degradation or chirality inversion reactions.

3. Dexamethasone C-C bond length varies from 1.34 to 1.6 Å which is not good practice for structure determination and also have large ellipsoids.

The dexamethasone molecule in question comes from the model of a low occupancy site (Guest X: occupancy: 18.9%). Thus, the answer to this comment is tied to the responses to the first and second comments. To reiterate, low occupancy models are more strongly affected by the residual electron density from the disorder partners, which means that the bond lengths could exhibit a greater deviation from their expected values. That is why we provided comparison of bond lengths in each site and also to the pure molecular

crystals (Figs. S14-18), which helped to assess the correctness of each model and correlate it with occupancy. To clarify this discussion, the following changes were made to the main text:

Page 5, paragraph 2: “Comparison of bond lengths in both crystals showed that the differences were generally negligible, and the corresponding errors were in a similar range (Figs. S14-18). The more significant differences in some individual bonds could be explained by the differing surrounding environments and interactions since the guests inside Co-3TPHAP were also subjected to additional solvation effects.”

was changed to

“Comparison of bond lengths in both crystals showed that the differences were generally negligible, and the corresponding errors were in a similar range, especially when the guest occupancy was above 50% (Figs. S14-18). Since the lower occupancy (<50%) guest models were more strongly affected by residual electron density from the disorder partners, some bond lengths deviated significantly out of the normal range. Additionally, the more significant differences in some individual bonds could be explained by the differing surrounding environments and interactions since the guests inside Co-3TPHAP were also subjected to additional solvation effects.”

4. Points 1-3 are very commonly observed in crystalline sponge method and I am not convinced that how this sponge has better resolution than any other crystalline sponge published before. To make their argument better authors should have compared their finding with other published work.

The reviewer is correct to point out the common drawbacks that occur in the crystallographic analysis of previously reported crystalline sponges. However, when evaluating our work, the reviewer decided to focus almost exclusively on the deficiencies of structure models of low occupancy guests and using them to seemingly dismiss the overall structure analysis. As mentioned in the responses to earlier comments, every single organic molecule in the current manuscript had at least one high occupancy site which could be refined down to atomic level resolution. From the perspective of future users of the crystalline sponge technique (natural product chemists, medicinal chemists, etc.), this outcome can be considered a success since the goal of obtaining the high-quality structure of a target molecules have been achieved. This is the key difference between Co-3TPHAP and other reported crystalline sponges, which typically only provide a molecular-level resolution for the structures of encapsulated guests regardless of the number of sites or their occupancy.

As stated in a seminal review on this topic (*Angew. Chem. Int. Ed.* **60**, 25204–25222 (2021)), “The chemical structures can be confirmed from the analysis, but precise structural parameters such as bond lengths, angles, and hybridization manner should not be discussed because the data obtained by the CS method are generally of lower resolution”. This is the consequence of observing a molecule only at a molecular-level resolution. In contrast, we were able to confidently discuss the structural parameters of the encapsulated guest molecules with high accuracy. Another important outcome of this work is the unique mechanism that facilitated the structure analysis down to the atomic level resolution. That is the presence of highly malleable hydrogen-bonded water networks inside Co-3TPHAP that uniquely adapted to maximize the interactions with each guests and align them inside the pore. In most of previously reported crystalline sponges the encapsulated molecules interacted with the host framework primarily through weak secondary interactions, which did not effectively affix their position and orientation causing their electron densities to become more diffuse. In a few instances, coordination to the open metal sites was used as a relatively strong highly directional interaction mode, however, these cannot be universally applicable. As such, the formation of a complex set of interactions between the host framework, the guest and internal solvent, leading to an improved crystallographic resolution is a unique feature of Co-3TPHAP.

Another important consideration when analyzing the crystalline sponge data is how the disordered low occupancy electron densities, that are commonly observed inside porous frameworks, are dealt with. One of the standard procedures is to remove these electron densities altogether using the PLATON SQUEEZE

command, which can significantly simplify the refinement process. In fact, this command has been used during the crystalline sponge analysis in the past, which is one of the strategies for lowering the R factors (*Nature* **501**, 262–262 (2013)). However, the developer of PLATON SQUEEZE stated in the past (*Acta Crystallogr. Sect. C* **71**, 9–18 (2015)); “Using SQUEEZE as part of the MOF soaking method (Inokuma et al., 2013), where the interest lies in the guest region as opposed to the host region, can be very challenging, is not recommended and should be done with extreme care when attempted”. In the case of Co-3TPHAP, well resolved diffraction spots could be detected, even in the higher angle region, which made it possible to assign almost the entirety of the pore space, and as a consequence, visualize multiple guest sites. In the process, we completely avoided the use of PLATON SQUEEZE command, thus further underscoring the reliability of our structure models.

Based on the combination of these factors, we firmly believe that the structure analysis of organic molecules by encapsulation inside Co-3TPHAP presented in this manuscript is superior to the current standard of crystalline sponge structures and sets a new benchmark in this field.

5. There were several MOFs were used as successful crystalline sponges authors mention in the manuscript ‘ To date only a handful of MOFs have been identified to be compatible for a wide range of substrates’ but did not cite other crystalline sponges.

To address this comment, several citations that highlight previously used MOFs were added to the quoted sentence (the citation numbers are shown as they appear in the manuscript):

2. Inokuma, Y. et al. X-ray analysis on the nanogram to microgram scale using porous complexes. *Nature* **495**, 461–466 (2013).
 3. Zigon, N., Duplan, V., Wada, N. & Fujita, M. Crystalline Sponge Method: X-ray Structure Analysis of Small Molecules by Post-Orientation within Porous Crystals—Principle and Proof-of-Concept Studies. *Angew. Chem. Int. Ed.* **60**, 25204–25222 (2021).
 15. Tu, T. N. & Scheer, M. A novel crystalline template for the structural determination of flexible chain compounds of nanoscale length. *Chem* **9**, 227–241 (2023).
 25. Lee, S., Kapustin, E. A. & Yaghi, O. M. Coordinative alignment of molecules in chiral metal-organic frameworks. *Science* **353**, 808–811 (2016).
 26. de Poel, W. et al. The Crystalline Sponge Method in Water. *Chem. Eur. J.* **25**, 14999–15003 (2019).
 27. Mon, M. et al. Crystallographic snapshots of host-guest interactions in drugs@metal-organic frameworks: Towards mimicking molecular recognition processes. *Mater. Horiz.* **5**, 683–690 (2018).
6. Authors did not cite reviews written on crystalline sponge methods.

ACS cent. Sci., (2021), **7**, 404-414.

Angew.Chem.Int.Ed., (2021), **60**, 25204–25222.

Materials Today: Proceedings, (2022), **56**, 3766-3773.

We believe that we provided sufficient references to properly cover the background of the crystalline sponge field. In particular, the second paper on this list have already been cited in the manuscript as number 3.

Reviewer #3 (Remarks to the Author):

The authors have obtained a series of interesting crystal structures of host-guest combinations in this paper, but the ideal is not very novel, and the strategy used is also the strategy of many published papers before.

Personally, this is actually a very classical host-guest chemistry, through the crystallization method to obtain some host-guest combined crystal structure, not suitable for the concept of atomic resolution. There is a suspicion of grandstanding. It is suggested that the author should readjust his thinking and condense the innovative ideas of the paper. So I do not recommend publishing in this journal with the present manuscript.

The reviewer is indeed correct in saying that crystal structures presented in this manuscript can be categorized as classical host-guest chemistry. However, we would like to point out that the more important questions are what kind of new information could be obtained from our host-guest structures and if this information is of high value. In the current work, we demonstrated the use of Co-3TPHAP framework as a crystalline sponge to solve the structures of a wide variety of pharmaceutically relevant molecules, three of them for the first time.

Regarding the term “atomic level resolution”, we believe that its usage in the manuscript is fully appropriate based on the provided results. Figure R2 in the response to reviewer 2 highlights the difference between models with atomic (from the present work) and molecular level (from a structure solved using different crystalline sponge) resolutions. The former has clearly defined localized electron densities that are perfectly superimposed over the expected guest molecules. This level of resolution permitted us to assign atoms with very minimal ambiguity. On the other hand, in the latter model, the electron densities are more diffuse and only roughly fit the outline of the molecule. Because of that, a significant number of restraints and constraints have to be applied to atom positions and atomic displacement parameters in order to construct a reasonable model. Every guest molecule presented in this work (Figure 1 in the manuscript) was refined down to atomic level resolution (as consistent with the above definition) for at least one crystallographic site. Most of the crystalline sponges reported in the past, such as the popular ZnI₂-TPT framework originally reported by Fujita, cannot claim such a level of structure quality for the majority of encapsulated guests. This outcome is significant because it places Co-3TPHAP as one of the best performing crystalline sponge materials reported to date that is applicable for a wide range of guest chemistries. In that sense, this manuscript succeeded in advancing the development of crystalline sponges, especially considering the relative paucity of MOFs in the literature that are suitable for this application. Furthermore, it has a potential to have a broad impact in the field of structure determination of complex organic molecules.

It is recommended that the authors provide the powder diffractions of these host-guest compounds to demonstrate that a large amount of samples can be obtained.

The powder diffraction can be useful for determination of phase purity of the sample and for monitoring any changes in the structure that accompany guest encapsulation. However, considering the complexity of the presented host-guest structures, the amount of useful structural information that can be extracted using this technique is limited. In general, single crystal diffraction analysis, in this context, would provide superior results. To illustrate, we measured PXRD of the artemether@Co-3TPHAP sample (the encapsulation was performed in *n*-heptane at 40 °C for 3 days) and compared it to the simulated pattern, as well as the starting framework (Figure R3). The data shows changes to some diffraction peaks, notably around the 5° and 8°, suggesting lowering of the crystal symmetry. These results confirmed the artemether inclusion into the pore of Co-3TPHAP. However, the measured pattern did not perfectly match the simulated pattern, which could be due to the differences in experimental conditions between powder and single crystal, such as temperature

and the presence of solvent. Overall, no significant new information regarding the host-guest structure was obtained from this experiment and therefore inclusion of this data to the manuscript was deemed not essential.

Figure R3. Powder X-ray diffraction patterns of Co-3TPHAP before and after artemether encapsulation. Simulation from the single crystal diffraction analysis of artemether@Co-3TPHAP (black), Co-3TPHAP containing artemether (blue), and the starting Co-3TPHAP (red). The patterns were collected on a Rigaku Miniflex powder diffractometer with D/teX Ultra (1D) detector using Cu $K\alpha$ radiation at room temperature. The samples were placed on the silicon plate as suspensions in *n*-heptane and covered with Mylar film to prevent solvent evaporation during measurements.

Regarding the amount of Co-3TPHAP that can be obtained. The crystalline sponge technique requires only a few crystals for the analysis of one compound. In addition, each crystal was checked by the single crystal X-ray diffraction, which means that the structure of Co-3TPHAP was confirmed each time. As such, it was not necessary to synthesize large amounts of framework for this study and confirm its phase purity by PXRD.

At the same time, there are some writing errors in the article, such as the use of italics in the writing of space groups. Please provide the revised documents.

We apologize for the presence of basic errors. Based on this comment, the manuscript was carefully checked and revised accordingly:

page 2, paragraph 5 and page 3, paragraph 1: “ $P2_1/n$ ” was changed to “ $P2_1/n$ ”

page 3, paragraph 1: “ C_2/c ” was changed to “ C_2/c ”

Supplementary material, page 28: “ $P2_1/c$ ” was changed to “ $P2_1/c$ ”

Additional changes made to the manuscript:

Main text, page 3, paragraph 2: the beginning sentence “In the case of artemether, the encapsulated crystal displayed a chiral C_2 space group with the Flack parameter of 1.9(2)%.” was moved to the end of the preceding paragraph.

Main text, page 4, paragraph 2, the first sentence: “Fig. S4-5” was changed to “Figs. S4-5”.

Main text, page 4, paragraph 3, the second to last sentence: “Fig. 8- S10” was changed to “Figs. S8-10”.

Main text, page 6, paragraph 5, the second to last sentence: “highly direction hydrogen bonds” was changed to “highly directional hydrogen bonds”.

Main text, page 14, acknowledgments: “Grant Number JPMJSP2106JSPS, KAKENHI” was changed to “Grant Number JPMJSP2106, JSPS KAKENHI”.

SI, Figure S15: the legend was missing “%” in “Guest Y” and “Guest X” parts. The figure was updated to include the missing sign.

Reviewers' Comments:

Reviewer #1:

Remarks to the Author:

Authors have addressed all my concerns.

While concerning the mismatch of PXRD in Figure R3, the explanation is not convincing enough. Different experimental conditions could cause some difference but it could also be due to some other reasons. I think better data need be collected and then at least Le Bail fitting needs be done to confirm that all peaks can be assigned to certain HKL with slightly changed unit cell.

Reviewer #2:

Remarks to the Author:

Comments to authors.

Authors has now provided the reasonable explanation for their isotropic refinement and lower occupancies of their guest molecules. These issues are very common with the Crystalline Sponge Method and therefore reasonable explanation is required in the manuscript. I can now recommend this version of the paper for publication in Nature Communication.

Reviewer #3:

Remarks to the Author:

The author answered two of the three questions, but as for the powder diffraction of the sample, I personally think the author did not give a good response, so I do not recommend the paper to be published in Nature communication.

Response to the Second Round of Reviewer's Comments for the Manuscript NCOMMS-23-25075-A:

Reviewer/Editor Comments to the Authors are in *black*, and our responses are in *blue*.

REVIEWER COMMENTS

Reviewer #1 (Remarks to the Author):

Authors have addressed all my concerns.

While concerning the mismatch of PXRD in Figure R3, the explanation is not convincing enough. Different experimental conditions could cause some difference but it could also be due to some other reasons. I think better data need to be collected and then at least Le Bail fitting needs to be done to confirm that all peaks can be assigned to certain HKL with slightly changed unit cell.

It is difficult to obtain better diffraction data due to the sensitivity of Co-3TPHAP to changes in solvation environment. The MOF crystals have to be kept immersed in a solvent at all times, otherwise their crystallinity rapidly degrades. Because of that, the data will always contain a background scattering from the *n*-heptane solvent and the covering Mylar film. Nevertheless, to address the reviewer's comment, we performed the Le Bail refinement on the previously measured PXRD pattern of Artemether@Co-3TPHAP (Figure R3 in the first response letter). A reasonable fit (Figure R4) was obtained using the same space group as that in the single crystal structure. The refined unit cell parameters were slightly different to those for the single crystal at 95 K (Table R1). Furthermore, all the observed diffraction peaks aligned well with the allowed reflections for the *C*2 space group, as highlighted by the blue tick marks. This result indicates that due to its soft flexible nature, the MOF structure was modified in response to the change of measurement conditions between single crystal and powder (temperature, solvation environment), thus validating our earlier explanation.

The aim of the crystal sponge method is to determine molecular structures utilizing the pores within a MOF single crystal. It is crucial to emphasize that our research deviates from most of other MOF studies, such as gas sorption or separation. We would appreciate understanding the goals of our research.

Figure R4 Le Bail refinement of the Artemether@Co-3TPHAP PXRD pattern measured in *n*-heptane, as shown in Figure R3 in the first response letter. Measured pattern (black crosses), Le Bail fit (red solid line), residual (green solid line) and allowed reflections for the *C2* space group at Cu $K\alpha_1$ and $K\alpha_2$ radiation wavelengths (blue tick marks). The refinement was performed using Rietica.

Table R1 The unit cell parameters for Artemether@Co-3TPHAP obtained from the SXRD and PXRD measurements.

	Space group	a / Å	b / Å	c / Å	β / °	Volume / Å ³
SXRD (95 K)	C2	31.781(3)	17.143(1)	29.748(2)	118.776(4)	14205.9
PXRD (300 K)	C2	32.2(2)	17.5(1)	30.1(1)	119.5(2)	14762

Reviewer #2 (Remarks to the Author):

Comments to authors.

Authors has now provided the reasonable explanation for their isotropic refinement and lower occupancies of their guest molecules. These issues are very common with the Crystalline Sponge Method and therefore reasonable explanation is required in the manuscript.

I can now recommend this version of the paper for publication in Nature Communication.

We thank the reviewer for their recommendation.

Reviewer #3 (Remarks to the Author):

The author answered two of the three questions, but as for the powder diffraction of the sample, I personally think the author did not give a good response, so I do not recommend the paper to be published in Nature communication.

Regarding the PXRD of the encapsulated Co-3TPHAP, we fully addressed the issue in the response to the first reviewer.

Reviewers' Comments:

Reviewer #1:

Remarks to the Author:

To get a better data is not that difficult for this sample. You need just a better instrument. The peak width, I/σ etc. can be significantly improved. With a new unit cell, the new simulated PXRD can be put below the experimental PXRD for clear comparison.

I strongly recommend the author to collect a better data.

Reviewer/Editor Comments to the Authors are in *black*, and our responses are in *blue*.

REVIEWER COMMENTS

Reviewer #1 (Remarks to the Author):

To get a better data is not that difficult for this sample. You need just a better instrument. The peak width, I/σ etc. can be significantly improved. With a new unit cell, the new simulated PXRD can be put below the experimental PXRD for clear comparison.

I strongly recommend the author to collect a better data.

Better powder diffraction data of Artemether@Co-3TPHAP at room temperature was collected using Rigaku SmartLab diffractometer. The MOF crystals immersed in *n*-heptane were loaded into a borosilicate glass capillary (diameter = 0.5 mm). The open end of the capillary was sealed with clay to prevent solvent evaporation during the measurements.

The resultant powder pattern had better resolution and signal-to-noise ratio compared to the previous one (Figure R4 in the second response letter) revealing weaker diffraction peaks. Le Bail refinement of the data showed that the *C2* space group could provide a reasonable fit to most of the observed diffraction peaks (Figure R5). The obtained unit cell parameters were slightly different to the single crystal structure.

Figure R5: Powder pattern of Artemether@Co-3TPHAP (black) and the corresponding Le Bail fit (red). The inset shows the fitted unit cell parameters.

We would like to acknowledge Dr. Kiyohiro Adachi and Dr. Daisuke Hashizume (RIKEN), as well as Xiaohan Wang and Prof. Yoichi Murakami (Tokyo Tech) for their help with PXRD measurements.

Reviewers' Comments:

Reviewer #1:

Remarks to the Author:

The result looks fine now.